# Edible Far Eastern Ferns as a Dietary Source of Long-Chain Polyunsaturated Fatty Acids

**DOI:** 10.3390/foods10061220

**Published:** 2021-05-28

**Authors:** Eduard V. Nekrasov, Vasily I. Svetashev

**Affiliations:** 1Amur Branch, Botanical Garden-Institute of the Far Eastern Branch of the Russian Academy of Sciences, Blagoveshchensk, 675000 Amur Oblast, Russia; 2National Scientific Center of Marine Biology of the Far Eastern Branch of the Russian Academy of Sciences, 690024 Vladivostok, Russia; vsvetashev@mail.ru

**Keywords:** *Matteuccia struthiopteris*, *Osmundastrum asiaticum*, *Pteridium aquilinum*, ostrich fern, bracken fern, long-chain polyunsaturated fatty acids, arachidonic acid, eicosapentaenoic acid

## Abstract

Young fronds of ferns are consumed as a vegetable in many countries. The aim of this study was to analyze three fern species that are available for sale in the Russian Far East as dietary sources in terms of fatty acids that are important for human physiology: arachidonic acid (20:4n-6, ARA), eicosapentaenoic acid (20:5n-3, EPA) and other valuable long-chain polyunsaturated fatty acids. The content of ARA and EPA was 5.5 and 0.5 mg/g dry weight, respectively, in *Pteridium aquilinum*, 4.1 and 1.1 in *Matteuccia struthiopteris*, and 2.2 and 0.8 in *Osmundastrum asiaticum*. Salted fronds of *P. aquilinum* contained less these fatty acids than the raw fronds, with a decrease of up to 49% for ARA and 65% for EPA. These losses were less pronounced or even insignificant in dried fronds. Cooked ferns preserved significant portions of the long-chain polyunsaturated fatty acids: cooked *P. aquilinum* contained 4.4 mg/g dry weight ARA and 0.3 mg/g dry weight EPA. The ferns may provide a supplemental dietary source of these valuable long-chain polyunsaturated fatty acids, especially for vegetarian diets.

## 1. Introduction

Ferns belong to a group of free-sporing vascular plants (class Polypodiopsida) which is wide-spread around the world in different climatic zones. The fern sporophyte is usually the dominant life stage (generation) both in terms of size and lifespan. The sporophyte typically has conspicuous leaves (fronds), which are compound and display a great diversity in size and shape. The global economic value of ferns is much lower than that of seed plants; however, there are a number of examples of the ethnobotanical usage of ferns [1]. The traditional and modern usage of ferns for food has been observed in different regions [1,2,3,4,5]. Fronds (mostly young), rhizomes, stems and tubers can be eaten as fresh greens, cooked as vegetables or processed for starch [4].

The most popular edible ferns available on the market include the bracken fern (*Pteridium aquilinum*) and the ostrich fern (*Matteuccia struthiopteris*). The ostrich fern is more popular in eastern North America [3]. According to an earlier estimate, sales of the ostrich fern fiddleheads might be in the range of USD 7–10 million in North America [3]. Edible ferns available on the market in Japan are the bracken fern, the ostrich fern and *Osmunda japonica*, which are priced at JPY 1.2 million (ca. USD 11,000), JPY 5.4 million (ca. USD 49,500), and JPY 22.1 million (ca. USD 202,700) per ton, respectively [6]. Two ferns (*P. aquilinum* and *O. japonica*) have been reported for South Korean markets [7]. Furthermore, in China, where about 52 species of ferns are consumed as food, fronds on the market are mostly represented by *P. aquilinum* var. *latiusculum* and *O. japonica*, with the bracken fern occupying more than 80% of the fern food market [4]. The bracken fern is also the most harvested fern across Siberia and the Far East of Russia. In the three regions of the Russian Far East (Khabarovsky Krai, Primorsky Krai, and Sakhalin Oblast), its total annual harvest reached 2500 ton in 1976–1980 due to its high demand in Japan [8]. Subsequently, the export of the salted fern shoots decreased; however, it may still be an essential part of the local economy. According to customs data, the Far Eastern region of Russia exported 94.1 tons of ferns in 2016–2019.

The young fern fronds can be preserved in different ways: by freezing and canning (in North America), salting (mostly in Russia) and drying (in China and East Asia) [3,4,8]. 

There are at least 13 edible fern species in the Russian Far East [2]. Unlike the most harvested bracken fern, which is widely sold and seemingly may be found on a shelf in any supermarket in the region, the ostrich fern is less common for the local market. However, the ostrich fern is promoted as being safer than other edible ferns [9]. The fern *Osmunda japonica*, which is popular in East Asia, grows in Russia only on Sakhalin Island. Instead, *Osmundastrum asiaticum* is used for food consumption [2] and is also consumed in China (referred to as *Osmundastrum cinnamomeum* in [4]). 

Fern green tissue is of particular interest due to its fatty acid composition. Unlike flowering plants, fern fronds contain long-chain polyunsaturated fatty acids (LC-PUFAs)—arachidonic acid (20:4n-6, ARA), eicosapentaenoic acid (20:5n-3, EPA), sciadonic acid (5,11,14-20:3, SCA), juniperonic acid (5,11,14,17-20:4, JA) and other acids [10]. LC-PUFAs are considered to be valuable nutrients due to their physiological roles and limited biosynthesis in humans. The most common and important among them are ARA, EPA, and docosahexaenoic acid (22:6n-3, DHA) [11,12,13]. LC-PUFAs are precursors of different signaling molecules and are important for the regulation of membrane properties [13]. These fatty acids and their metabolites are involved in normal and pathological processes, ranging from inflammation and its resolution to the functioning of the nervous system [13]. LC-PUFAs have to be obtained from food, and they also can be synthesized endogenously from the corresponding essential fatty acids (18:2n-6 and 18:3n-3). However, both dietary sources and the endogenous conversion appear to be important for normal metabolism [13].

The objective of this study was to analyze the edible ferns grown in the Russian Far East as dietary sources of LC-PUFAs. The hypothesis behind this study was that commercially available products of these ferns, which are processed using methods traditional for the East Asian region, preserve the valuable fatty acids. The obtained results aim at the establishment of ferns as a plant source of LC-PUFAs, and are thus of interest in the formulation of new vegetarian diets containing these fatty acids.

## 2. Materials and Methods

### 2.1. Plant Material and Chemicals

Raw young fronds were of two origins: from Primorsky Krai and from Amur Oblast, Russia. The fronds of Primorsky origin were purchased at a local market in Vladivostok (*Pteridium aquilinum* (L.) Kuhn and *Matteuccia struthiopteris* (L.) Tod.) or harvested in the surroundings of Vladivostok (*Osmundastrum asiaticum* Tagawa) in May 2018. The fronds of Amur origin (*P. aquilinum* and *M. struthiopteris*) were harvested from plants growing in the collection of genetic resources of plants of the Amur Branch of the Botanical Garden-Institute FEB RAS or nearby (Blagoveshchensk, Amur Oblast, Russia) in May 2018. The fronds of *O. asiaticum* were harvested in Blagoveshchensky District in May 2019. Voucher specimens of *P. aquilinum* and *O. asiaticum* (mature fronds) were deposited in the Herbarium of the Botanical Garden-Institute (VBGI) under accession numbers ABGI53085–ABGI53086 and ABGI78410, respectively. Dried ostrich fern was purchased from a private seller. According to the seller, the ostrich fern was harvested in the Smirnykhovsky District of Sakhalin Oblast (the central part of Sakhalin Island) in 2019; the fiddleheads were boiled in water for 5–7 min, followed by soaking in cold water and drying in the sun. Dried bracken ferns were purchased from an online shop (http://shop.soyka.ru/ (accessed on 7 July 2019)) in 2019. The product was produced by Food Company INS E (Dunin, China) in April 2018, with a shelf-life of 24 months. Salted bracken ferns were produced by two different companies from Amur Oblast (IP Onishko E.V., Blagoveshchensk, Russia) and Khabarovsky Krai (Lesnye produkty Ltd., Khabarovsk, Russia) and purchased from local supermarkets in 2018. Cooked bracken fern, under the market name “Spicy fern salad”, was produced by Intel Ltd. (Blagoveshchensk, Russia) and bought from a local supermarket. According to the product information, the composition of the salad was as follows: fern fiddleheads, vegetable oil, garlic, salt, glutamate, coriander, red pepper, sorbic acid. An appetizer with Osmunda fern (named in the menu as Osmund’s fiddlehead fern starter) was obtained from the Port Cafe restaurant (Vladivostok, Russia). According to the chef, the dried fern used to prepare the appetizer was purchased under the name of Osmunda from a private seller at a local market. Only petioles of young fronds without crosiers were found in the dish.

Freshly harvested fronds were divided into crosiers with about 1 cm of an underlying petiole segment and boiled in water for 3 min. The material is referred to below as ‘raw fiddleheads/fronds’. A portion of the boiled crosiers of *O. asiaticum* were dried at room temperature under exposure to sunlight for 6 days. The salted bracken fern was soaked in tap water overnight, with two changes of water. The fronds were blotted with paper tissue. Crosiers with about 1-cm-long petiole segments were taken for analysis. Pieces of crosiers of the cooked bracken fern from the salad were washed with warm tap water and blotted with paper tissue. The cooked Osmunda fern from the appetizer was prepared for lipid extraction in a similar way. The plant material was used immediately for lipid extraction or stored at −20 °C until extraction. For the dry weight determination, plant material was dried at 110 °C until reaching a constant weight.

Chloroform, methanol, benzene, hexane, and sulfuric acid were of analytical grade (Vekton, Saint Petersburg, Russia). Chloroform, hexane, and methanol were distilled before use.

### 2.2. Lipid Extraction

Lipids were extracted with chloroform–methanol, as described previously [10]. The dried fronds were ground with a mortar and pestle first and soaked in distilled water at a volume based on the dry weight determined for the freshly harvested fronds of the corresponding species. Methanol (2.5 *v*/*wt*) and chloroform (1.25 *v*/*wt*) were added on the basis of the total weight of the dry material and the added water. After grinding, the liquid portion of the mixture was transferred onto a cotton wool filter. The residue was repeatedly extracted with chloroform (2.5 *v*/*wt*) and methanol (1.25 *v*/*wt*) and transferred onto the filter. The procedure was repeated twice. The final residue on the filter was rinsed with chloroform–methanol (1:1, *v*/*v*) twice. The combined crude extract was partitioned with 0.9% aqueous NaCl in the ratio 2:1 (*v*/*v*). The chloroform layer was filtered through a cotton wool filter and evaporated on a rotary evaporator. The obtained total lipids were dissolved in chloroform–methanol (2:1, *v*/*v*) and stored in a freezer until analysis. The wet fronds (freshly harvested, salted and cooked) were extracted in a similar way, with the solvents used in volumes based on their corresponding wet weights.

### 2.3. Fatty Acid Analysis

Fatty acids were analyzed as methyl esters by means of gas-liquid chromatography with flame ionization detection (GLC-FID) and gas chromatography–mass spectrometry (GC-MS). Fatty acid methyl esters (FAMEs) were obtained as follows [14]: 2–4 mg of total lipids in 0.1 mL of benzene were mixed with 1 mL of 2% sulfuric acid in methanol (*wt*/*v*), the vial was flushed with nitrogen or argon and incubated at 50 °C overnight. After the addition of water (0.125 mL), FAMEs were extracted with hexane (0.5 mL and, repeatedly, 0.2 mL). The hexane layer was evaporated under reduced pressure and re-dissolved in 0.2 mL hexane–benzene (1:1, *v*/*v*). For fatty acid identification, FAMEs were purified by means of thin-layer chromatography on silica gel plates in benzene. For the quantitative analysis of fatty acid content (weight content in plant material), tricosanoic acid (23:0, Sigma-Aldrich, St. Louis, MO, USA) was added as an internal standard before the methanolysis. 4,4-dimethyloxazoline (DMOX) derivatives of fatty acids were prepared from FAMEs according to [15].

FAMEs were analyzed on a Shimadzu GC 2010 (Shimadzu Corporation, Kyoto, Japan) or Agilent 6890N (Agilent Technologies, Santa Clara, CA, USA) gas chromatograph, equipped with a Supelcowax 10 capillary column (30 m × 0.25 mm, 0.25 µm film thickness, Supelco, Bellefonte, PA, USA). The analysis conditions on the Shimadzu GC 2010 chromatograph were as follows: the carrier gas (helium) pressure was 19 psi; the oven temperature was 205 °C for 60 min. The chromatograph worked under Shimadzu GCsolution application (GCMSsolution ver. 4.11, Shimadzu Corporation, Kyoto, Japan). The analysis conditions on the Agilent 6890N chromatograph were: the carrier gas (helium) pressure was 21.8 psi; the initial temperature for the oven was 195 °C for 60 min, then 20 °C/min up to 250 °C and held for 50 min. The FAME solutions were injected manually at a volume of 1 μL. The chromatograph worked under ChemStation Rus software (InterLab Inc., 2005, Ver. A.10.02[1757], v.4.45 under license of Agilent Technologies).

GC-MS analyses were performed on a Shimadzu GCMS QP5050A apparatus (Shimadzu Corporation, Kyoto, Japan) as described previously [10,15]: the fused quartz capillary column Supelco MDN-5s (30 m × 0.25 mm I.D., Supelco, Bellefonte, PA, USA) was used and the carrier gas was helium. For FAMEs, the temperature program was from 160 °C up to 260 °C at the rate of 2 °C/min and then held for 30 min. For the DMOX derivatives, the temperature program was from 190 °C up to 270 °C at the rate of 2 °C/min and held for 30 min. Mass spectra were recorded at 70 eV. The chromatograph worked under Shimadzu GCMSsolution application (GCMSsolution ver. 4.11, Shimadzu Corporation, Kyoto, Japan).

The identification of fatty acids was performed as described in [10]. Palmitic (16:0) and stearic (18:0) acids used for the calculation of equivalent chain-lengths (ECLs) were obtained from Sigma-Aldrich (St. Louis, MO, USA). The quantification of individual fatty acids was based on their FID peak area relative to the internal standard (23:0). No correction factors were used in the calculation of fatty acid contents. Shorthand designations of unsaturated fatty acids are as follows: for the fatty acids with methylene-interrupted double bonds in the *cis* configuration, X:Yn-Z indicates a fatty acid chain of X carbon atoms with Y methylene-interrupted double bonds, where a terminal double bond is located at Z carbon atoms from the methyl terminus (n); a double bond in the *trans* configuration is indicated with “*t*-” in front of the number of carbon atoms; for Δ5-unsaturated polymethylene-interrupted fatty acids (sciadonic and juniperonic acids), the positions of all double bonds are indicated in front of the numbers of carbon atoms.

### 2.4. Statistical Analysis

Analysis of the data was performed using Microsoft Office Excel 2003 (Microsoft Corp., Redmond, WA, USA, 2003). In the Tables, results are shown as means for three samples ± standard deviation. *p*-values were calculated using the two-tailed unpaired Student’s *t*-test. *p* < 0.05 was considered statistically significant.

## 3. Results and Discussion 

### 3.1. Raw Young Fronds

The contents of total lipids and fatty acids in the raw fiddleheads of the edible ferns are shown in Table 1 and Appendix A. The fern species belong to different families: Onocleaceae (*Matteuccia struthiopteris*), Dennstaedtiaceae (*Pteridium aquilinum*) and Osmundaceae (*Osmundastrum asiaticum*). For each species, fronds were from two Russian Far Eastern regions—Primorsky Krai and Amur Oblast—with the collection sites being about 850 km apart. The content of total lipids ranged from 0.8–0.9% of wet weight in *O. asiaticum* to 1.0–1.2% in *P. aquilinum*. The lipid content in the fiddleheads of *P. aquilinum* was significantly higher (*p* < 0.05) than in the other two species. The lipid content was higher in the ferns harvested in Amur Oblast; however, the difference between the two regions was significant only for *P. aquilinum* (Appendix A).

Fatty acids of the young fronds included saturated (14:0, 15:0, 16:0, 17:0, 18:0, 20:0, 22:0, 24:0, 26:0), monoenoic (16:1n-9, 16:1n-7, *t*-16:1n-13, 18:1n-9, 18:1n-7, 18:1n-5, 20:1n-9), polyenoic (16:2n-6, 18:2n-6, 20:2n-6, 16:3n-3, 18:3n-6, 18:3n-3, 18:4n-3, 20:3n-6, 20:4n-6, 20:5n-3) and Δ5-unsaturated polymethylene-interrupted (5,11,14-20:3 and 5,11,14,17-20:4) acids (Table 1). Traces of 23:0, 16:1n-5 and 20:3n-3 were also detected in some samples (Appendix A). The major fatty acids of the fiddleheads were palmitic 16:0 (25.0–30.0% of total fatty acids), linoleic 18:2n-6 (17.0–27.0%), α-linolenic 18:3n-3 (10.8–24.0%), ARA 20:4n-6 (6.4–13.5%), and oleic 18:1n-9 (5.0–10.0%) acids. Significant portions were found for 24:0 (1.3–3.2%), EPA 20:5n-3 (0.8–3.2%), 16:3n-3 (1.3–2.8%), γ-linolenic 18:3n-6 (0.9–2.7%) and dihomo-γ-linolenic 20:3n-6 (0.9–2.6%) acids, 18:0 (1.0–2.0%), 22:0 (0.6–2.5%) and 18:1n-7 (0.6–1.6%). All other fatty acids were usually below 1% (Table 1). Although the fern species did not differ significantly in terms of the sum of polyunsaturated fatty acids (PUFAs), they were different in terms of the content of omega-6 (n-6) and omega-3 (n-3) fatty acids. 

*P. aquilinum* had the highest ratio of n-6/n-3 fatty acids (2.6–3.2) due to the high levels of linoleic acid and, to a lesser extent, ARA, and lower percentages of α-linolenic acid and EPA. In contrast, *O. asiaticum* was found to have equal percentages of n-6 and n-3 fatty acids, with the highest contents of α-linolenic, 16:3n-3, 18:4n-3, EPA and JA among the species. *M. struthiopteris* contained intermediate levels of linoleic and α-linolenic acids, with relatively high percentages of ARA and EPA (Table 1). Other differences among the species include higher contents of minor saturated fatty acids (18:0, 20:0, 22:0 and 24:0) in *P. aquilinum*, and some minor monounsaturated (16:1n-9, 16:1n-7, 18:1n-7) and polyunsaturated (20:3n-6) fatty acids in *M. struthiopteris*.

Regional differences in the fatty acid percentages were not similar for the fern species. The sum of PUFAs was significantly higher in *M. struthiopteris* from Primorsky Krai and *O. asiaticum* from Amur Oblast, whereas there was no difference between the two regions for *P. aquilinum* (Appendix A). However, the sum of n-6 fatty acids were higher in *P. aquilinum* and *M. struthiopteris* from Primorsky Krai, without significant differences in the sum of n-3 fatty acids between the ferns collected in the two regions. Among the major fatty acids, regional differences were found for 18:2n-6 in *P. aquilinum*, 18:1n-9 in *M. struthiopteris*, 16:0 in *O. asiaticum* and ARA for all three species. There were significant differences in the percentages of some minor fatty acids in the ferns from the two regions (Appendix A).

In the literature, data on the fatty acid composition of young fronds of edible ferns cover *M. struthiopteris* of North American origin [16,17], *P. aquilinum* from Russia [18,19], as well as these and other species of the European origin [20]. In general, the contents of fatty acids in the young fronds of *P. aquilinum* and *M. struthiopteris* have a similar tendency: high percentages of palmitic and linoleic acids, followed by α-linolenic and/or ARA, which have been found previously for these [17,18] and other fern species [10]. *O. asiaticum* is an exception, since its fiddleheads contained more α-linolenic acid than linoleic acid, significantly less ARA and more EPA than other species (Table 1). A very high content of γ-linolenic acid (18:3n-6) was found for fern fiddleheads in a previous study [20], which was comparable or even exceeded the content of palmitic acid. This was not observed in our study (Table 1) or in other studies [17,18,19].

### 3.2. Processed Fronds

Processed fronds (dried or salted) were not significantly different from freshly harvested fronds of corresponding species in terms of lipid content on the dry weight basis (Table 2, Table 3 and Table 4). A significant difference was found only for the salted fronds of the bracken fern from Amur Oblast; however, the value was closer to the freshly harvested fronds (7.9% vs. 8.4% dry weight) as compared to the two other products (7.7% and 7.8%), which had slightly higher deviations of their means (Table 2).

When commercially available fern fronds processed in different ways were compared in terms of their fatty acid content (Table 2 and Table 3), the biggest difference was noted for salted fronds of *P. aquilinum*. The salted products from two different producers were characterized by higher percentages of saturated fatty acids (mostly due to palmitic acid), monounsaturated fatty acids (due to oleic acid), and a lower one of polyunsaturated fatty acids (both n-6 and n-3) as compared to the freshly harvested fronds (Table 2). The difference in the weight content of fatty acids was less pronounced for saturated and monounsaturated fatty acids, but was reproduced for PUFAs. All the major PUFAs, including ARA and EPA, were significantly lower in the salted fronds.

The dried bracken fern product from China contained less total lipids, total fatty acids and individual fatty acids on a dry weight basis than the raw fronds harvested in Amur Oblast; however, the differences were often less pronounced than in the case of the salted products (Table 2). Moreover, the raw fronds and the dried product were quite similar in their percentages of fatty acids (Table 2). Significant differences were found only for saturated (more palmitic acid in the dried fronds), monounsaturated (less oleic acid in the dried fronds), and some minor PUFAs (18:3n-6, 18:4n-3, 20:2n-6, SCA). Similar results were observed for the dried ostrich fern product from Sakhalin Island and the raw fronds of *M. struthiopteris* harvested in Amur Oblast—the samples contained similar levels of total lipids (the content was insignificantly higher in the dried fronds), as well as total and individual fatty acids (Table 3). The dried fronds from Sakhalin Island contained more α-linolenic acid and less oleic acid (both in percentage and weight content). Other differences were related to some minor fatty acids.

Although the drying method was known for the ostrich fern from Sakhalin Oblast (see Section 2. Materials and Methods), it was unknown for the bracken fern product from China. To investigate the effect of drying on fatty acid content in fern fronds, fatty acids were analyzed in fiddleheads of *O. asiaticum* dried under two conditions: at 110 °C for 26 h or at room temperature under exposure to sunlight for 6 days (Table 4). The dried fiddleheads were found to contain less total fatty acids, as well some major fatty acids (16:0, 18:2n-6, ARA) than the raw fronds; however, the differences were insignificant. The percentage of fatty acids was almost identical for the raw and dried fiddleheads, except for ARA, of which the percentage was slightly lower (*p* < 0.01) after drying at 110 °C (Table 4). These results indicate that the drying method has little effect on the fatty acid content in fern fronds—they can be dried quickly at a high temperature or slowly at a low temperature (providing that the fronds are preliminary boiled to inactivate enzymes—see below).

Fern fronds processed in different ways have been analyzed previously in terms of their nutrient composition: canned and frozen ostrich fern fiddleheads [21], salted (with and without preliminary blanching) ostrich fern fiddleheads [9], ostrich fern fiddleheads stored in cold water [22] and frozen and salted bracken fern fronds [19]. A small decrease in fat (lipid) content in the canned and frozen fiddleheads was observed, as compared to raw ones, accompanied by increased water content [21]. The storage of raw ostrich fern fiddleheads in cold water led to the accumulation of some fatty acids, including saturated, mono- and polyunsaturated acids (but not EPA), an effect that was attributed to cold temperature acclimation and caused by fatty acid upregulation [22]. Preliminary thermal treatment (boiling in water for 1 min) of ostrich fern fiddleheads had a positive effect on the quality and shelf-life of the salted product and was suggested as a preferable method for enzyme inactivation [9].

Enzyme activity may be a reason for the differences observed in the fatty acid content of the salted bracken fern fronds (Table 2), since dry salting of freshly harvested fronds is commonly used for their storage. Quick enzyme inactivation after boiling in water or heating at a high temperature during the drying process provides better preservation of PUFAs than the protracted process of salting, even at a high salt concentration. The influence of salt on lipid oxidation in meat and seafood products is under discussion and possible mechanisms of salt pro-oxidant activity have been suggested [23]—disruption of cell membrane integrity, liberation of iron ions and inhibition of antioxidant enzymes. Such processes can also occur during blanching or freezing of fern fronds. However, boiling in water with subsequent soaking in cold water and drying in the sun, performed on the ostrich fern fiddleheads, resulted in a fatty acid content similar to that of freshly harvested plant material (Table 3). A high level of PUFAs (with ARA up to 14.8% vs. 24.9% of palmitic acid) was found in frozen fronds of the bracken fern [19]. Although enzymes responsible for lipid degradation during the salting of fern fronds are yet to be identified, blanching/boiling of fern fronds before salting may improve the content of PUFAs in the salted product.

### 3.3. Cooked Ferns

There are different dishes with ferns in the Russian Far East. Usually, fern fronds are boiled, then fried or stewed and used in a hot dish with meat, in salads or appetizers. The purpose of this part of the study was to confirm the preservation of the valuable PUFAs (ARA, EPA, γ-linolenic and dihomo-γ-linolenic acids) in cooked ferns. Since meat contains all the fatty acids, we analyzed dishes with ferns that were free of any meat: a salad with the bracken fern and an appetizer with Osmunda fern (see Section 2. Materials and Methods).

The percentages of fatty acids in the cooked ferns were quite different from all other samples (raw and processed) due to the high levels of oleic and linoleic acids (Table 5). The major portion of the fatty acids obviously originated from vegetable oils used for the frying or dressing of the ferns. Indeed, the content of total lipids in the parts of the bracken fern from the salad was almost twice as high as that measured in its raw fronds in wet and dry weights (Table 2 and Table 5). The weight content of oleic and linoleic acids in the fern from the salad also significantly exceeded their content in the uncooked fronds (more than six times higher than in the fronds of the salted bracken fern from Amur Oblast). The weight contents of the valuable PUFAs (ARA, EPA, γ-linolenic, and dihomo-γ-linolenic acids), as well as some major fatty acids (palmitic and α-linolenic), were close to those of the salted fern from Amur Oblast, although there were significant differences compared to those of the fern from Khabarovsky Krai. The ARA/EPA ratio was also close to that observed for the salted fern from Amur Oblast (14.4 vs. 14.7 for the salted product from Amur Oblast, Table 2 and Table 5).

In contrast, the fern in the appetizer was not significantly different from the crosiers of *O. asiaticum* in terms of the total lipid content (Table 4 and Table 5). Although the percentages for oleic and linoleic acids were also twice as high than those observed in the raw fiddleheads, their weight contents (along with palmitic acid) were not different between the cooked and raw material. The majority of other fatty acids, including the polyunsaturated ones of interest, were significantly lower in weight content in the cooked fern parts (up to 7.5 times for ARA). A likely reason for this observation may be the parts of the fern used in the dish. The appetizer was found to contain only pieces of frond petioles, which can be depleted in lipids as compared to leafy fiddleheads. Although we did not separately analyze the petioles of the raw fronds of *O. asiaticum*, the ARA to EPA ratio was close to the values found for the fern fiddleheads (2.2 vs. 2.7–2.8, Table 4 and Table 5). Though lipid losses in the fern parts during cooking cannot be ruled out, the petioles of *P. aquilinum* contained about an order of magnitude less polar lipids than the pinnae of this species [24].

Altogether, these results demonstrate the preservation of the valuable PUFAs in processed and cooked ferns; thus, fatty acids become part of a diet when ferns are consumed. These results are in agreement with the literature data. Changes in the fatty acid composition of food during cooking have been the subject of a number of studies [25,26,27,28,29,30]. There is accumulation of the oxidized products of PUFAs under cooking conditions [30]. However, the changes depend on the cooking method. For fish enriched in PUFAs, frying has the highest impact on fatty acid profiles, not only due to the direct degradation of PUFAs but also because of the composition of the frying oil absorbed by the product [26,27,28,29]. When the absolute content of the fatty acids is considered, cooking often does not significantly reduce the levels of the valuable LC-PUFAs [25,26,28,29,30].

### 3.4. Edible Ferns in Comparison with Other Dietary Sources of LC-PUFAs

LC-PUFAs are considered important for human physiology and have bioactive potential [13,31,32]. Fish is the major source of long-chain omega-3 fatty acids (EPA, DHA), whereas long-chain omega-6 fatty acids (ARA and its precursor, 20:3n-6) are consumed mainly with meat and other foods of animal origin [11].

The ARA intake for normal healthy adults was estimated to be 100–250 mg/day in advanced countries, whereas vegetarians may consume much less: 3–44 mg/day [11]. The edible ferns in our study were found to contain ARA: 80.1 mg/100 g wet weight (5.5 mg/g dry weight) for *P. aquilinum*, 52.5 mg/100 g wet weight (4.1 mg/g dry weight) for *M. struthiopteris* and 40.2 mg/100 g wet weight (2.15 mg/g dry weight) for *O. asiaticum* (Table 2, Table 3 and Table 4 and Appendix A). In the cooked ferns, the ARA content was found to be 72.0 ± 4.3 mg/100 g wet weight for the bracken fern from the salad and 7.1 ± 1.9 mg/100 g wet weight for the Osmunda fern from the appetizer (results are not shown). Thus, the edible ferns contain ARA in quantities comparable with meat (4.8–97 mg/100 g wet weight) [26,33,34,35,36,37], many fish species (0.29–0.85 mg/g dry weight, 30–212 mg/100 g wet weight) [25,30,38] and edible macroalgae (0.13–5.4 mg/g dry weight) [39,40].

The average daily intake of EPA is 33 mg in the USA, though there are significant differences depending on age, gender and pregnancy status [12]. EPA is a minor fatty acid in the edible ferns evaluated here (Table 1, Table 2, Table 3, Table 4 and Table 5), of which the content hardly exceeded 1 mg/g dry weight (Table 3). On the wet weight basis, its content in the ferns was: 7.1 mg/100 g for *P. aquilinum*, 14.1 mg/100 g for *M. struthiopteris* and 14.5 mg/100 g for *O. asiaticum* (Appendix A). In terms of EPA content, the ferns are obviously inferior to fish (3.1–7.2 mg/g dry weight and up to 1.7 g/100 g wet weight) [25,27,28,29,38], some macroalgae (0.39–8.3 mg/g dry weight) [39,40] and some meat products (29, 28, 27, and 46 mg/100 g wet weight for beef, veal, lamb and mutton, respectively [33], 21–33 mg/100 g wet weight for lamb [36]). However, the ferns are comparable in terms of EPA content with other kinds of meat, such as pork (2.2–5.3 mg/100 g wet weight [26,35]), rabbit (2.9–4.7 mg/100 g wet weight [34]) and broiler chicken (1.4–2.4 mg/100 g wet weight [37]).

Unlike animal sources, terrestrial ferns do not contain DHA [10,20]. However, the edible ferns contain precursors of ARA and EPA which can contribute to their dietary value. The content of γ-linolenic acid (18:3n-6) and 18:4n-3 reached 0.5–1.2 and 0.1 mg/g dry weight, respectively, in the ferns (Table 2, Table 3 and Table 4). Fish has been found to contain less γ-linolenic acid (0–0.2 mg/g dry weight) and more 18:4n-3 (0.1–0.7 mg/g dry weight) [25] compared to these results. Similar or lower values were found for dihomo-γ-linolenic acid (20:3n-6) and 20:4n-3 in the ferns (Table 2, Table 3 and Table 4), as has also been shown for fish [25].

Potentially bioactive [32] Δ5-unsaturated polymethylene-interrupted fatty acids, such as 5,11,14-20:3 and 5,11,14,17-20:4, are minor components in the young fronds of the edible ferns (Table 1) with the maximum content of only 0.2 and 0.1 mg/g dry weight (or 0.8% and 0.5% of total fatty acids) for SCA and JA, respectively, in *O. asiaticum* (Table 4). Seeds of some conifer and flowering species contain much more these fatty acids [41,42] though not all of them are usually consumed as food.

Here, we have demonstrated that Far Eastern edible ferns may serve as raw material for a vegetarian product line with different ARA/EPA (and omega-6/omega-3 fatty acid) ratios: ranging from 11–17 (omega-6/omega-3 = 2.6–3.2) in *P. aquilinum* to 2–3 (omega-6/omega-3 = 1.0) in *O. asiaticum*. Advances in industrial lipid extraction from plant material may promote the emergence of such products derived from these ferns. A method for the production of lipid-containing fractions was recently described for fern fronds, using dimethyl ether and its mixture with water and ethanol under near-critical conditions [43]. This process is more environmentally friendly and technologically suitable for the scaled-up extraction of plant lipids than with the use of convenient liquid organic solvents. The application of green technology has opened up opportunities for the development of new products from these and other edible ferns.

In general, this study has proven the dietary value of edible ferns from the Russian Far East as sources of LC-PUFAs. However, the processing of these ferns’ young fronds should be improved in order to avoid the loss of these acids, as was found with the fiddleheads after dry salting.

## 4. Conclusions

(1)Fiddleheads of three fern species, harvested in the Russian Far East for food consumption, contain the valuable omega-6 and omega-3 long-chain polyunsaturated fatty acids (arachidonic acid, eicosapentaenoic acid, γ-linolenic acid, dihomo-γ-linolenic acids, 20:4n-3) in amounts comparable to certain meats, as well as minor quantities of Δ5-unsaturated polymethylene-interrupted fatty acids (sciadonic and juniperonic acids).(2)Arachidonic acid always prevailed over eicosapentaenoic acid in terms of its content in the studied species. The highest ratio of omega-6/omega-3 fatty acids, the highest level of ARA and the lowest level of EPA were observed in *Pteridium aquilinum*. The lowest ratio of omega-6/omega-3 fatty acids, along with the highest percentage of EPA and the lowest one of ARA, among the species was found in *Osmundastrum asiaticum*, whereas *Matteuccia struthiopteris* contained a high percentage of ARA, along with a relatively high content of EPA and an intermediate ratio of omega-6/omega-3.(3)Plant material from fern species collected from two separate Far Eastern regions was different in terms of the percentage of ARA in the total fatty acids, although the ratios of ARA to EPA were similar for each species.(4)Salted fronds were much more different from freshly harvested fronds in terms of the content of long-chain polyunsaturated fatty acids, compared to that of dried frond samples. Prompt heat treatment (boiling, blanching, drying) of the freshly harvested fronds may be the preferable method for the preservation of long-chain polyunsaturated fatty acids in the processed material.(5)Valuable long-chain polyunsaturated fatty acids are well preserved in the cooked fern fronds, thus becoming a part of the diet when ferns are included.

## Figures and Tables

**Table 1 foods-10-01220-t001:** Lipid and fatty acid content in the raw fiddleheads of edible ferns obtained from the Russian Far East.

	*Pteridium aquilinum*		*Matteuccia struthiopteris*		*Osmundastrum asiaticum*	
	Primorsky Krai	Amur Oblast	Primorsky Krai	Amur Oblast	Primorsky Krai	Amur Oblast
Total lipids, % wet weight						
	1.02 ± 0.07	1.23 ± 0.10	0.85 ± 0.12	0.99 ± 0.14	0.80 ± 0.09	0.92 ± 0.06
Fatty acid, % of sum						
14:0	0.3 ± 0.1	0.3 ± 0.1	0.3 ± 0.1	0.3 ± 0.0	0.1 ± 0.0	0.1 ± 0.0
14:1n-5	0.2 ± 0.0	0.1 ± 0.0	0.1 ± 0.0	0.2 ± 0.0	0.2 ± 0.0	0.1 ± 0.0
15:0	0.1 ± 0.0	0.1 ± 0.0	0.3 ± 0.0	0.2 ± 0.0	0.1 ± 0.0	0.1 ± 0.0
16:0	26.3 ± 0.9	25.0 ± 0.8	26.6 ± 0.6	26.6 ± 1.0	30.0 ± 0.3	25.3 ± 0.5
16:1n-9	0.1 ± 0.0	0.1 ± 0.0	0.5 ± 0.0	0.8 ± 0.0	0.2 ± 0.0	0.1 ± 0.0
16:1n-7	0.2 ± 0.0	0.3 ± 0.1	0.9 ± 0.0	1.2 ± 0.1	0.2 ± 0.0	0.2 ± 0.0
*t*-16:1n-13	0.1 ± 0.1	0.2 ± 0.1	0.2 ± 0.0	0.3 ± 0.1	0.2 ± 0.1	0.1 ± 0.0
16:2n-6	0.1 ± 0.1	0.1 ± 0.1	0.2 ± 0.1	0.3 ± 0.0	0.1 ± 0.1	0.1 ± 0.0
16:3n-3	1.7 ± 0.2	1.9 ± 0.6	1.3 ± 0.1	1.7 ± 0.4	2.5 ± 0.3	2.8 ± 0.2
17:0	0.1 ± 0.0	0.1 ± 0.0	0.1 ± 0.0	0.2 ± 0.0	0.1 ± 0.0	0.1 ± 0.0
18:0	1.7 ± 0.0	2.0 ± 0.2	1.0 ± 0.0	1.5 ± 0.1	1.6 ± 0.5	1.2 ± 0.0
18:1n-9	5.0 ± 0.2	5.9 ± 0.4	5.5 ± 0.3	10.0 ± 0.1	5.9 ± 0.5	5.8 ± 0.2
18:1n-7	0.6 ± 0.0	0.9 ± 0.2	1.5 ± 0.0	1.6 ± 0.2	1.0 ± 0.1	1.0 ± 0.3
18:1n-5	0.1 ± 0.0	0.1 ± 0.0	0.2 ± 0.0	0.1 ± 0.0	0.3 ± 0.1	0.3 ± 0.0
18:2n-6	27.0 ± 0.4	25.1 ± 0.8	20.9 ± 0.5	19.9 ± 0.7	18.0 ± 1.6	17.0 ± 1.0
18:3n-6	0.9 ± 0.0	2.5 ± 0.1	1.7 ± 0.1	2.7 ± 0.1	2.2 ± 0.3	2.4 ± 0.1
18:3n-3	10.8 ± 0.6	12.8 ± 1.3	16.9 ± 0.1	14.4 ± 1.1	22.0 ± 1.3	24.0 ± 1.2
18:4n-3	0.1 ± 0.1	0.2 ± 0.0	0.1 ± 0.0	0.2 ± 0.0	0.5 ± 0.1	0.6 ± 0.1
20:0	1.4 ± 0.1	1.5 ± 0.2	0.3 ± 0.0	0.4 ± 0.0	0.5 ± 0.2	0.4 ± 0.0
20:1n-9	0.1 ± 0.0	0.1 ± 0.0	0.2 ± 0.0	0.2 ± 0.0	0.1 ± 0.0	0.1 ± 0.0
20:2n-6	0.2 ± 0.0	0.1 ± 0.0	0.3 ± 0.0	0.1 ± 0.0	0.2 ± 0.0	0.2 ± 0.0
5,11,14-20:3	0.5 ± 0.0	0.4 ± 0.0	0.2 ± 0.0	0.2 ± 0.0	0.5 ± 0.0	0.8 ± 0.1
20:3n-6	1.3 ± 0.1	1.5 ± 0.4	2.6 ± 0.2	1.8 ± 0.1	0.9 ± 0.1	1.3 ± 0.1
20:4n-6 (ARA)	13.5 ± 0.4	11.8 ± 0.4	12.8 ± 0.5	9.3 ± 0.1	6.4 ± 0.5	8.8 ± 0.2
5,11,14,17-20:4	0.1 ± 0.0	0.1 ± 0.0	0.2 ± 0.0	0.2 ± 0.0	0.4 ± 0.0	0.5 ± 0.0
20:4n-3	0.1 ± 0.0	0.1 ± 0.0	0.2 ± 0.0	0.1 ± 0.0	0.2 ± 0.0	0.3 ± 0.0
20:5n-3 (EPA)	0.8 ± 0.0	1.0 ± 0.2	2.3 ± 0.2	2.3 ± 0.2	2.9 ± 0.2	3.2 ± 0.3
22:0	2.5 ± 0.4	2.1 ± 0.2	0.6 ± 0.1	0.8 ± 0.1	0.9 ± 0.1	1.2 ± 0.0
23:0	0.2 ± 0.0	0.1 ± 0.1	0.1 ± 0.0	0.1 ± 0.1	0.2 ± 0.0	n.d.
24:0	3.2 ± 0.7	2.7 ± 0.2	1.5 ± 0.3	1.5 ± 0.1	1.3 ± 0.1	1.4 ± 0.0
26:0	0.6 ± 0.2	0.5 ± 0.0	0.5 ± 0.2	0.4 ± 0.1	0.3 ± 0.1	0.3 ± 0.1
Others ^1^	0.8 ± 0.1	0.7 ± 0.0	1.0 ± 0.1	1.1 ± 0.1	1.0 ± 0.1	0.8 ± 0.0
SFAs	36.5 ± 0.7	34.5 ± 1.6	31.3 ± 0.9	32.2 ± 1.2	35.1 ± 0.5	30.0 ± 0.6
MUFAs	6.4 ± 0.1	7.8 ± 0.3	9.1 ± 0.3	14.6 ± 0.2	8.1 ± 0.4	7.9 ± 0.2
PUFAs	57.1 ± 0.6	57.7 ± 1.3	59.6 ± 1.0	53.3 ± 1.4	56.8 ± 0.9	62.1 ± 0.8
n-6	43.5 ± 0.3	41.5 ± 0.8	38.7 ± 1.1	34.3 ± 0.5	28.3 ± 0.8	30.6 ± 1.4
n-3	13.6 ± 0.8	16.1 ± 1.9	20.9 ± 0.2	18.9 ± 1.8	28.5 ± 1.6	31.4 ± 1.5
ARA/EPA	17.0	11.3	5.6	3.9	2.2	2.8
(n-6)/(n-3)	3.2	2.6	1.8	1.8	1.0	1.0

^1^ Others include the minor fatty acids 14:1n-5, 15:0, 16:1n-5, 16:2n-6, 17:0, 18:1n-5, 20:3n-3 and 23:0. Abbreviations: ARA—arachidonic acid, EPA—eicosapentaenoic acid, MUFAs—monounsaturated fatty acids, n.d.—not determined, PUFAs—polyunsaturated fatty acids, SFAs—saturated fatty acids. Note: Values are means of three samples ± standard deviations.

**Table 2 foods-10-01220-t002:** Lipid and fatty acid content of raw and processed fiddleheads of bracken fern (*Pteridium aquilinum*).

	Raw	Dried	Salted 1	Salted 2	Raw	Dried	Salted 1	Salted 2
Total lipids								
% wet weight					1.23 ± 0.10 a,b	6.88 ± 0.84 c	0.98 ± 0.03 d	1.07 ± 0.07 b,d
% dry weight					8.43 ± 0.17 a	7.65 ± 0.96 a,b	7.90 ± 0.25 b	7.84 ± 0.52 a,b
Fatty acid	% of sum				mg/g dry weight			
14:0	0.3 ± 0.0 a	0.4 ± 0.1 a,c	0.5 ± 0.1 b,c	0.5 ± 0.1 b,c	0.15 ± 0.03 a	0.13 ± 0.05 a	0.20 ± 0.06 a	0.19 ± 0.03 a
16:0	24.3 ± 0.6 a	27.4 ± 0.9 b	36.2 ± 0.2 c	37.1 ± 0.5 c	11.07 ± 0.38 a,b	9.25 ± 1.78 b	15.08 ± 2.22 a,c	12.87 ± 0.49 b,c
16:1n-9	0.15 ± 0.02 a,b	0.12 ± 0.01 b	0.16 ± 0.00 a	0.17 ± 0.00 a	0.07 ± 0.01 a	0.04 ± 0.01 b	0.07 ± 0.01 a	0.06 ± 0.00 a
16:1n-7	0.3 ± 0.1 a	0.2 ± 0.0 a	0.4 ± 0.0 a	0.3 ± 0.0 a	0.12 ± 0.02 a	0.09 ± 0.03 a	0.15 ± 0.02 a	0.12 ± 0.01 a
*t*-16:1n-13	0.16 ± 0.04 a	0.16 ± 0.03 a	0.40 ± 0.01 b	0.37 ± 0.04 b	0.07 ± 0.02 a	0.06 ± 0.02 a	0.17 ± 0.03 b	0.13 ± 0.01 b
16:3n-3	2.0 ± 0.5 a,b	1.4 ± 0.1 b	1.0 ± 0.0 a	1.0 ± 0.1 a	0.92 ± 0.27 a	0.49 ± 0.11 a	0.44 ± 0.07 a	0.36 ± 0.03 a
18:0	1.9 ± 0.1 a	2.1 ± 0.0 a	2.5 ± 0.1 b	2.7 ± 0.1 c	0.88 ± 0.03 a,b	0.70 ± 0.11 a	1.06 ± 0.19 a,b	0.94 ± 0.05 b
18:1n-9	5.8 ± 0.5 a	4.4 ± 0.6 b	7.4 ± 0.1 c	7.2 ± 0.2 c	2.64 ± 0.35 a	1.51 ± 0.32 b	3.08 ± 0.52 a	2.51 ± 0.14 a
18:1n-7	0.9 ± 0.2 a,c	0.6 ± 0.0 b	0.9 ± 0.0 c	1.1 ± 0.1 c	0.39 ± 0.06 a	0.21 ± 0.05 b	0.38 ± 0.05 a	0.37 ± 0.03 a
18:2n-6	25.4 ± 0.9 a	24.4 ± 0.8 a	19.2 ± 0.3 b	21.2 ± 0.3 c	11.59 ± 0.89 a	8.30 ± 1.33 b	8.02 ± 1.16 b	7.33 ± 0.16 b
18:3n-6	2.6 ± 0.2 a	1.9 ± 0.2 b	1.6 ± 0.0 b	1.5 ± 0.0 b	1.16 ± 0.02 a	0.64 ± 0.15 b	0.66 ± 0.10 b	0.53 ± 0.02 b
18:3n-3	13.1 ± 1.0 a	11.4 ± 0.2 a	7.3 ± 0.1 b	6.6 ± 0.4 b	5.99 ± 0.73 a	3.91 ± 0.66 b	3.04 ± 0.49 b	2.30 ± 0.05 b
18:4n-3	0.20 ± 0.02 a	0.15 ± 0.01 b	0.11 ± 0.00 c	0.10 ± 0.00 c	0.09 ± 0.01 a	0.05 ± 0.01 b	0.04 ± 0.01 b	0.04 ± 0.00 b
20:0	1.4 ± 0.2 a	1.4 ± 0.1 a	1.6 ± 0.2 a	1.5 ± 0.0 a	0.63 ± 0.10 a	0.48 ± 0.10 a	0.67 ± 0.16 a	0.53 ± 0.02 a
20:1n-9	0.13 ± 0.03 a,b,c	0.14 ± 0.02 a,b	0.18 ± 0.01 a,c	0.18 ± 0.02 a,b,c	0.06 ± 0.02 a	0.05 ± 0.01 a	0.08 ± 0.02 a	0.06 ± 0.01 a
20:2n-6	0.13 ± 0.00 a	0.37 ± 0.02 b	0.25 ± 0.01 c	0.23 ± 0.02 c	0.06 ± 0.00 a	0.12 ± 0.03 a	0.10 ± 0.02 a	0.08 ± 0.01 a
5,11,14-20:3	0.35 ± 0.06 a	0.55 ± 0.07 b	0.37 ± 0.02 a	0.35 ± 0.06 a	0.16 ± 0.03 a	0.18 ± 0.04 a	0.15 ± 0.01 a	0.12 ± 0.02 a
20:3n-6	1.6 ± 0.5 a,b	1.7 ± 0.0 a	2.4 ± 0.1 b	1.7 ± 0.1 a	0.74 ± 0.23 a,b	0.60 ± 0.12 a	1.00 ± 0.11 b	0.59 ± 0.04 a
20:4n-6 (ARA)	12.1 ± 0.7 a	13.7 ± 1.4 a	9.2 ± 0.2 b	8.2 ± 0.2 c	5.51 ± 0.07 a	4.63 ± 0.77 a,b	3.85 ± 0.51 b	2.83 ± 0.08 b
5,11,14,17-20:4	0.09 ± 0.02 a	0.08 ± 0.01 a	0.14 ± 0.01 b	0.07 ± 0.01 a	0.04 ± 0.01 a,b	0.03 ± 0.01 a	0.06 ± 0.01 b	0.02 ± 0.00 a
20:4n-3	0.12 ± 0.02 a	0.12 ± 0.01 a	0.16 ± 0.01 b	0.10 ± 0.00 a	0.05 ± 0.01 a	0.04 ± 0.01 a	0.07 ± 0.01 a	0.03 ± 0.00 a
20:5n-3 (EPA)	1.1 ± 0.2 a	0.9 ± 0.1 a	0.6 ± 0.0 b	0.5 ± 0.0 c	0.49 ± 0.07 a	0.32 ± 0.05 b	0.26 ± 0.04 b,c	0.17 ± 0.00 c
22:0	2.0 ± 0.1 a	2.2 ± 0.1 a,b	2.2 ± 0.2 a,b	2.3 ± 0.1 b	0.91 ± 0.07 a	0.74 ± 0.14 a	0.92 ± 0.20 a	0.78 ± 0.02 a
24:0	2.7 ± 0.3 a	3.0 ± 0.3 a,b	3.5 ± 0.3 b	3.3 ± 0.1 a,b	1.22 ± 0.17 a	1.00 ± 0.17 a	1.46 ± 0.32 a	1.15 ± 0.05 a
26:0	0.5 ± 0.0 a	0.6 ± 0.1 a	0.8 ± 0.0 b	0.7 ± 0.0 a	0.23 ± 0.03 a	0.20 ± 0.02 a	0.32 ± 0.06 a	0.23 ± 0.01 a
Others ^1^	0.7 ± 0.1 a	0.6 ± 0.0 a	1.0 ± 0.0 b	0.9 ± 0.1 b	0.31 ± 0.06 a,b	0.21 ± 0.03 a	0.42 ± 0.07 b	0.33 ± 0.04 b
SFAs	34.8 ± 1.2 a	38.1 ± 1.1 b	48.1 ± 0.7 c	49.0 ± 0.5 c	16.07 ± 0.83 a,b	13.03 ± 2.42 a	20.23 ± 3.27 b	17.10 ± 0.66 a,b
MUFAs	7.6 ± 0.4 a	5.9 ± 0.7 b	9.9 ± 0.1 c	9.9 ± 0.2 c	3.48 ± 0.36 a	2.03 ± 0.42 b	4.13 ± 0.67 a	3.44 ± 0.17 a
PUFAs	57.6 ± 1.3 a	56.0 ± 0.7 a	42.0 ± 0.8 b	41.1 ± 0.5 b	26.03 ± 1.55 a	18.91 ± 2.96 b	17.38 ± 2.48 b	14.13 ± 0.24 b
n-6	42.3 ± 1.4 a	42.7 ± 0.8 a	33.1 ± 0.6 b	33.2 ± 0.3 b	19.28 ± 1.10 a	14.49 ± 2.23 b	13.81 ± 1.91 b	11.50 ± 0.29 b
n-3	15.3 ± 1.0 a	13.3 ± 0.8 a	8.8 ± 0.4 b	7.9 ± 0.2 c	6.75 ± 0.74 a	4.42 ± 0.74 b	3.57 ± 0.56 b	2.63 ± 0.05 b
Total					45.58 ± 2.69 a	33.97 ± 5.69 a,b	41.74 ± 6.41 a,b	34.67 ± 1.05 b
ARA/EPA	11.2	14.8	14.7	16.7				
(n-6)/(n-3)	2.8	3.2	3.8	4.2				

^1^ Others include the minor fatty acids 14:1n-5, 15:0, 16:1n-5, 16:2n-6, 17:0, 18:1n-5 and 20:3n-3. Notes: Raw—fronds freshly harvested in Amur Oblast and boiled for 3 min before lipid extraction. Dried—dried fronds produced in China. Salted 1—salted fronds produced by the company from Amur Oblast. Salted 2—salted fronds produced by the company from Khabarovsky Krai. Values followed by the same letter were not significantly different (*p* > 0.05). See Table 1 for abbreviations and other details.

**Table 3 foods-10-01220-t003:** Lipid and fatty acid content of raw and processed fiddleheads of the ostrich fern (*Matteuccia struthiopteris*).

	Raw	Dried	Raw	Dried
Total lipids				
% wet weight			0.99 ± 0.14 a	7.88 ± 0.45 b
% dry weight			7.64 ± 1.16 a	8.84 ± 0.49 a
Fatty acid	% of sum		mg/g dry weight	
14:0	0.31 ± 0.01 a	0.31 ± 0.03 a	0.12 ± 0.02 a	0.12 ± 0.01 a
16:0	23.8 ± 0.1 a	24.6 ± 0.8 a	9.05 ± 1.17 a	9.39 ± 0.41 a
16:1n-9	0.69 ± 0.02 a	0.58 ± 0.06 a	0.26 ± 0.04 a	0.22 ± 0.03 a
16:1n-7	1.03 ± 0.03 a	0.83 ± 0.06 b	0.39 ± 0.05 a	0.32 ± 0.03 a
*t*-16:1n-13	0.28 ± 0.13 a	0.18 ± 0.06 a	0.11 ± 0.07 a	0.07 ± 0.03 a
16:3n-3	1.89 ± 0.51 a	1.87 ± 0.26 a	0.74 ± 0.29 a	0.72 ± 0.12 a
18:0	1.39 ± 0.06 a	1.21 ± 0.05 b	0.53 ± 0.08 a	0.46 ± 0.03 a
18:1n-9	9.2 ± 0.1 a	6.1 ± 0.4 b	3.49 ± 0.49 a	2.34 ± 0.22 b
18:1n-7	1.41 ± 0.16 a	1.10 ± 0.21 a	0.53 ± 0.05 a	0.42 ± 0.07 a
18:2n-6	20.4 ± 1.0 a	19.5 ± 0.5 a	7.72 ± 0.62 a	7.45 ± 0.16 a
18:3n-6	2.9 ± 0.0 a	2.3 ± 0.5 a	1.12 ± 0.15 a	0.87 ± 0.21 a
18:3n-3	15.9 ± 0.8 a	19.2 ± 0.4 b	6.08 ± 1.07 a	7.32 ± 0.44 a
18:4n-3	0.20 ± 0.01 a	0.17 ± 0.04 a	0.08 ± 0.01 a	0.07 ± 0.02 a
20:0	0.40 ± 0.04 a	0.38 ± 0.03 a	0.15 ± 0.02 a	0.15 ± 0.01 a
20:1n-9	0.23 ± 0.02 a	0.23 ± 0.02 a	0.09 ± 0.02 a	0.09 ± 0.01 a
20:2n-6	0.15 ± 0.03 a	0.35 ± 0.03 b	0.06 ± 0.00 a	0.13 ± 0.01 b
5,11,14-20:3	0.20 ± 0.02 a	0.32 ± 0.02 b	0.07 ± 0.01 a	0.12 ± 0.00 b
20:3n-6	1.99 ± 0.1 a	2.44 ± 0.3 a	0.75 ± 0.06 a	0.93 ± 0.08 b
20:4n-6 (ARA)	10.7 ± 0.6 a	11.4 ± 0.5 a	4.06 ± 0.30 a	4.36 ± 0.02 a
5,11,14,17-20:4	0.20 ± 0.01 a	0.23 ± 0.02 a	0.08 ± 0.01 a	0.09 ± 0.01 a
20:4n-3	0.17 ± 0.01 a	0.17 ± 0.01 a	0.06 ± 0.01 a	0.07 ± 0.00 a
20:5n-3 (EPA)	2.9 ± 0.1 a	2.5 ± 0.3 a	1.09 ± 0.16 a	0.94 ± 0.11 a
22:0	0.79 ± 0.04 a	0.74 ± 0.04 a	0.30 ± 0.03 a	0.28 ± 0.01 a
24:0	1.41 ± 0.07 a	1.46 ± 0.09 a	0.54 ± 0.05 a	0.56 ± 0.03 a
26:0	0.45 ± 0.03 a	0.39 ± 0.04 a	0.17 ± 0.03 a	0.15 ± 0.01 a
Others ^1^	1.0 ± 0.1 a	1.4 ± 0.1 b	0.40 ± 0.09 a	0.55 ± 0.08 a
SFAs	29.8 ± 0.9 a	30.6 ± 1.4 a	11.65 ± 1.68 a	11.92 ± 0.53 a
MUFAs	13.2 ± 0.3 a	9.7 ± 0.4 b	5.03 ± 0.75 a	3.72 ± 0.27 a
PUFAs	57.0 ± 0.8 a	59.7 ± 1.2 b	21.36 ± 2.41 a	22.54 ± 0.81 a
n-6	36.7 ± 1.6 a	36.6 ± 0.9 a	13.89 ± 1.14 a	13.96 ± 0.30 a
n-3	20.3 ± 2.1 a	23.1 ± 1.6 a	7.46 ± 1.26 a	8.58 ± 0.52 a
Total			38.04 ± 4.84 a	38.19 ± 1.41 a
ARA/EPA	3.7	4.7		
(n-6)/(n-3)	1.8	1.6		

^1^ Others include the minor fatty acids 14:1n-5, 15:0, 16:1n-5, 16:2n-6, 17:0, 18:1n-5 and 20:3n-3. Notes: Raw—fresh fronds harvested in Amur Oblast and boiled for 3 min before lipid extraction. Dried—dried fronds produced in Sakhalin Oblast. See Table 1 and Table 2 for abbreviations and other details.

**Table 4 foods-10-01220-t004:** Lipid and fatty acid content of raw and processed fiddleheads of *Osmundastrum asiaticum*.

	Raw	Dried at 110 °C	Air-Dried	Raw	Dried at 110 °C	Air-Dried
Total lipids						
% wet weight				0.92 ± 0.06 a	1.03 ± 0.06 a	0.92 ± 0.05 a
% dry weight				4.96 ± 0.49 a	5.50 ± 0.24 a	4.95 ± 0.58 a
Fatty acid	% of sum			mg/g dry weight		
14:0	0.13 ± 0.01 a	0.12 ± 0.01 a	0.12 ± 0.01 a	0.03 ± 0.00 a	0.03 ± 0.00 b	0.03 ± 0.00 b
16:0	25.3 ± 0.5 a	26.0 ± 0.3 a	25.6 ± 0.4 a	6.2 ± 0.4 a	5.9 ± 0.2 a	5.6 ± 0.4 a
16:1n-9	0.13 ± 0.01 a	0.14 ± 0.00 a	0.14 ± 0.01 a	0.03 ± 0.00 a	0.03 ± 0.00 a	0.03 ± 0.00 a
16:1n-7	0.17 ± 0.03 a	0.19 ± 0.02 a	0.17 ± 0.03 a	0.04 ± 0.00 a	0.04 ± 0.00 a	0.04 ± 0.00 a
*t*-16:1n-13	0.06 ± 0.00 a	0.06 ± 0.02 a	0.06 ± 0.00 a	0.01 ± 0.00 a	0.01 ± 0.00 a	0.01 ± 0.00 a
16:3n-3	2.8 ± 0.2 a	2.8 ± 0.1 a	2.8 ± 0.1 a	0.69 ± 0.09 a	0.62 ± 0.05 a	0.60 ± 0.07 a
18:0	1.16 ± 0.05 a	1.19 ± 0.05 a	1.20 ± 0.05 a	0.28 ± 0.01 a	0.27 ± 0.02 a	0.26 ± 0.03 a
18:1n-9	5.8 ± 0.2 a	6.1 ± 0.3 a	5.9 ± 0.51 a	1.42 ± 0.15 a	1.38 ± 0.12 a	1.29 ± 0.23 a
18:1n-7	0.99 ± 0.30 a	0.99 ± 0.23 a	1.03 ± 0.32 a	0.24 ± 0.06 a	0.22 ± 0.04 a	0.22 ± 0.05 a
18:2n-6	17.0 ± 1.0 a	17.5 ± 0.6 a	16.7 ± 0.9 a	4.1 ± 0.5 a	3.9 ± 0.2 a	3.6 ± 0.4 a
18:3n-6	2.4 ± 0.1 a	2.4 ± 0.1 a	2.4 ± 0.04 a	0.58 ± 0.06 a	0.53 ± 0.04 a	0.52 ± 0.06 a
18:3n-3	24.0 ± 1.2 a	24.2 ± 0.9 a	24.2 ± 1.1 a	5.8 ± 0.5 a	5.5 ± 0.3 a	5.2 ± 0.5 a
18:4n-3	0.56 ± 0.05 a	0.56 ± 0.03 a	0.57 ± 0.05 a	0.14 ± 0.01 a	0.12 ± 0.01 a	0.12 ± 0.01 a
20:0	0.35 ± 0.02 a	0.33 ± 0.03 a	0.37 ± 0.03 a	0.09 ± 0.00 a	0.07 ± 0.01 a	0.08 ± 0.01 a
20:1n-9	0.13 ± 0.03 a	0.15 ± 0.03 a	0.16 ± 0.03 a	0.03 ± 0.01 a	0.03 ± 0.01 a	0.04 ± 0.01 a
20:2n-6	0.23 ± 0.02 a	0.22 ± 0.01 a	0.24 ± 0.03 a	0.06 ± 0.01 a	0.05 ± 0.01 a	0.05 ± 0.01 a
5,11,14-20:3	0.79 ± 0.07 a	0.72 ± 0.04 a	0.78 ± 0.07 a	0.19 ± 0.02 a	0.16 ± 0.01 a	0.17 ± 0.02 a
20:3n-6	1.29 ± 0.14 a	1.18 ± 0.07 a	1.26 ± 0.13 a	0.31 ± 0.05 a	0.27 ± 0.02 a	0.27 ± 0.05 a
20:4n-6 (ARA)	8.9 ± 0.2 a	7.8 ± 0.1 b	8.6 ± 0.2 a	2.15 ± 0.21 a	1.76 ± 0.07 a	1.86 ± 0.18 a
5,11,14,17-20:4	0.47 ± 0.01 a	0.46 ± 0.01 a	0.49 ± 0.04 a	0.12 ± 0.01 a	0.10 ± 0.01 a	0.11 ± 0.02 a
20:4n-3	0.31 ± 0.01 a	0.29 ± 0.01 a	0.29 ± 0.01 a	0.08 ± 0.01 a	0.07 ± 0.01 a	0.06 ± 0.01 a
20:5n-3 (EPA)	3.2 ± 0.3 a	2.9 ± 0.2 a	3.2 ± 0.3 a	0.78 ± 0.10 a	0.66 ± 0.06 a	0.69 ± 0.09 a
22:0	1.17 ± 0.04 a	1.12 ± 0.08 a	1.18 ± 0.05 a	0.29 ± 0.03 a	0.25 ± 0.02 a	0.26 ± 0.03 a
24:0	1.39 ± 0.03 a	1.27 ± 0.06 a,b	1.31 ± 0.04 b	0.34 ± 0.03 a	0.29 ± 0.02 b	0.28 ± 0.02 b
26:0	0.29 ± 0.07 a	0.27 ± 0.08 a	0.25 ± 0.06 a	0.07 ± 0.01 a	0.06 ± 0.02 a	0.05 ± 0.01 a
Others ^1^	1.0 ± 0.1 a	1.0 ± 0.1 a	1.0 ± 0.0 a	0.24 ± 0.00 a	0.23 ± 0.02 a	0.23 ± 0.02 a
SFAs	31.8 ± 2.0 a	33.2 ± 0.4 a	32.9 ± 0.4 a	8.0 ± 0.5 a	7.5 ± 0.2 a	7.1 ± 0.6 a
MUFAs	7.9 ± 0.2 a	8.3 ± 0.1 a	8.1 ± 0.2 a	1.9 ± 0.1 a	1.9 ± 0.1 a	1.8 ± 0.2 a
PUFAs	60.4 ± 2.2 a	58.6 ± 0.4 a	59.0 ± 0.3 a	14.5 ± 1.2 a	13.2 ± 0.6 a	12.8 ± 1.2 a
n-6	30.7 ± 1.4 a	29.9 ± 0.7 a	30.1 ± 1.2 a	7.5 ± 0.8 a	6.7 ± 0.3 a	6.5 ± 0.7 a
n-3	29.7 ± 1.9 a	28.7 ± 0.9 a	29.0 ± 1.4 a	7.0 ± 0.6 a	6.5 ± 0.4 a	6.3 ± 0.6 a
Total				24.4 ± 1.8 a	22.5 ± 0.9 a	21.7 ± 2.0 a
ARA/EPA	2.8	2.7	2.7			
(n-6)/(n-3)	1.0	1.0	1.0			

^1^ Others include the minor fatty acids 14:1n-5, 15:0, 16:1n-5, 16:2n-6, 17:0, 18:1n-5 and 20:3n-3. See Table 1 and Table 2 for abbreviations and other details.

**Table 5 foods-10-01220-t005:** Lipid and fatty acid content of the cooked ferns.

	Bracken Fern (Salad)		Osmunda (Appetizer)	
Total lipids				
% wet weight		2.34 ± 0.27 ^§#^		0.98 ± 0.27
% dry weight		14.18 ± 1.61 ^§#^		3.94 ± 1.07
Fatty acid	% of sum	mg/g dry weight	% of sum	mg/g dry weight
14:0	0.19 ± 0.01 ^§#^	0.19 ± 0.02	0.17 ± 0.02 *	0.030 ± 0.007
16:0	14. 5 ± 0.9 ^§#^	14.7 ± 0.9	29.6 ± 5.6	5.33 ± 0.72
16:1n-9	0.07 ± 0.01 ^§#^	0.07 ± 0.00 ^#^	0.09 ± 0.01 *	0.016 ± 0.003 *
16:1n-7	0.18 ± 0.00 ^#^	0.18 ± 0.02 ^#^	0.14 ± 0.01	0.025 ± 0.007 *
*t*-16:1n-13	0.08 ± 0.02 ^#^	0.08 ± 0.02 ^§#^	0.07 ± 0.02	0.012 ± 0.001 *
16:3n-3	0.39 ± 0.02 ^§#^	0.39 ± 0.04	0.43 ± 0.06 *	0.079 ± 0.019 *
18:0	3.05 ± 0.05 ^§#^	3.12 ± 0.42 ^§#^	3.11 ± 0.13 *	0.589 ± 0.229
18:1n-9	20.5 ± 0.6 ^§#^	21.0 ± 3.2§ ^#^	13.0 ± 1.5 *	2.51 ± 1.18
18:1n-7	0.81 ± 0.03 ^#^	0.83 ± 0.13 ^§#^	1.33 ± 0.03	0.250 ± 0.091
18:2n-6	47.3 ± 1.1 ^§#^	48.4 ± 6.9 ^§#^	34.5 ± 5.4 *	6.71 ± 3.47
18:3n-6	0.75 ± 0.07 ^§#^	0.76 ± 0.08 ^#^	0.49 ± 0.03 *	0.093 ± 0.030 *
18:3n-3	3.10 ± 0.12 ^§#^	3.16 ± 0.34 ^#^	9.7 ± 0.1 *	1.81 ± 0.64 *
18:4n-3	0.05 ± 0.01 ^§#^	0.05 ± 0.01	0.12 ± 0.03 *	0.021 ± 0.002 *
20:0	0.55 ± 0.01 ^§#^	0.56 ± 0.06	0.44 ± 0.05	0.081 ± 0.018
20:1n-9	0.17 ± 0.00	0.17 ± 0.02 ^§#^	0.32 ± 0.04 *	0.059 ± 0.013
20:2n-6	0.08 ± 0.01 ^§#^	0.08 ± 0.01	0.55 ± 0.13 *	0.097 ± 0.009 *
5,11,14-20:3	0.14 ± 0.04 ^§#^	0.14 ± 0.02	0.27 ± 0.05 *	0.049 ± 0.010 *
20:3n-6	0.71 ± 0.09 ^#^	0.72 ± 0.03 ^§#^	0.60 ± 0.12*	0.110 ± 0.027 *
20:4n-6 (ARA)	4.3 ± 0.3 ^§#^	4.4 ± 0.3 ^#^	1.55 ± 0.21 *	0.286 ± 0.075 *
5,11,14,17-20:4	0.04 ± 0.01 ^§#^	0.04 ± 0.01 ^§^	0.14 ± 0.03 *	0.026 ± 0.005 *
20:4n-3	0.05 ± 0.01 ^§#^	0.05 ± 0.01 ^#^	0.18 ± 0.03 *	0.033 ± 0.009 *
20:5n-3 (EPA)	0.30 ± 0.01 ^§#^	0.30 ± 0.03 ^#^	0.71 ± 0.10 *	0.130 ± 0.031 *
22:0	1.15 ± 0.03 ^§#^	1.17 ± 0.11 ^#^	0.81 ± 0.13 *	0.146 ± 0.025 *
24:0	1.11 ± 0.10 ^§#^	1.13 ± 0.04	0.92 ± 0.28	0.161 ± 0.007 *
26:0	0.21 ± 0.03 ^§#^	0.21 ± 0.01	0.13 ± 0.04 *	0.023 ± 0.002 *
Others ^1^	0.30 ± 0.01 ^§#^	0.30 ± 0.03	0.61 ± 0.09 *	0.110 ± 0.021 *
SFAs	21.0 ± 1.2 ^§#^	21.5 ± 1.6 ^#^	35.7 ± 6.0	6.46 ± 1.01
MUFAs	21.9 ± 0.6 ^§#^	22.4 ± 3.3 ^§#^	15.1 ± 1.4 *	2.90 ± 1.30
PUFAs	57.1 ± 0.7 ^#^	58.2 ± 7.6 ^§#^	49.3 ± 4.7 *	9.46 ± 4.29
n-6	53.3 ± 0.6 ^§#^	54.5 ± 7.2 ^§#^	38.0 ± 5.0	7.36 ± 3.61
n-3	3.7 ± 0.1 ^§#^	3.7 ± 0.4 ^#^	11.3 ± 0.4 *	2.11 ± 0.69 *
Total		102.1 ± 12.4 ^§#^		18.82 ± 6.58
ARA/EPA	14.4		2.2	

^1^ Others include the minor fatty acids 14:1n-5, 15:0, 16:1n-5, 16:2n-6, 17:0, 18:1n-5 and 20:3n-3. Notes: The labels § and # indicate a significant difference (*p* ≤ 0.05) between the content of total lipids or fatty acids in the bracken fern from the salad and the salted bracken fern from Amur Oblast (§) or from Khabarovsky Krai (#), as shown in Table 2. The label * indicates a significant difference (*p* ≤ 0.05) between the content of total lipids or fatty acids in the Osmunda from the appetizer and the raw fiddleheads, as shown in Table 4. See Table 1 and Table 2 for abbreviations and other details.

## Data Availability

Data is contained within this article and Appendix A.

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
