# Peer review of "Edible Far Eastern Ferns as a Dietary Source of Long-Chain Polyunsaturated Fatty Acids"

_foods, 2021, doi:10.3390/foods10061220_

Round 1

Reviewer 1 Report

The presented manuscript is very interesting and touches an important subject of using ferns as a valuable source of long chain fatty acids. The authors also included in their work an important issue of preserving/loosing these valuable nutrients while using different food processing techniques.

The introduction part explains well the economic and nutritional importance of three dominating edible ferns in Russia and justifies the significance of presented study. The authors used adequate analytical methods. The data support the results.

I have few suggestions and questions:

  1. In the abstract, authors expressed the content of polyunsaturated fatty acids using diverse units (in mg/g, or in mg/100g) with respect to water content in plant mass. After reading the whole text, it make sense, however after first reading of the abstract only, it sounds a bit confusing. I suggest, though do not insist on the reconsideration using the units in the abstract.
  2. In chapter 2.3 the authors refers to their previous work on method description of identification and quantification of fatty acids. The ref. no 10 however, does not describe what standards were used for quantification of particular fatty acids and where the standards were obtained from. Please complete this information.
  3. In chapter Results and discussion, second paragraph, authors names individually those fatty acids that were found in studied raw material in amount equal or higher to 1%. Among them a vaccenic acid (11-octadecanoic acid 18:1n-7) is named. This fatty acid is naturally occurring as far as known in human milk, in ruminant’s fat and in dairy products. Can you discuss the presence of vaccenic acid in plant matrix since it is quite unique observation?
  4. The chapter number 3.1 is doubled – please renumber subchapters in Results and Discussion.
  5. Row 226, the authors cite findings on fatty acid content in ferns from ref. no 20. The results in this article with ref no 20 shows that the highest content was found for linoleic acid (18:2n-6). Out of the sentence in the row 226 it sounds that the highest content in cited ref was found for gamma-linoleic acid (18:3n-6) which is misleading. Please, reformulate the sentence.
  6. Chapter 3.3 – for better understanding of findings in fatty acid content in cooked ferns, would be reasonable to provide information on used frying oils, the amount and composition. Otherwise, it is not clear on what basis was performed the interpretation and comparison of fatty acid content in cooked and raw ferns. Is there for example any particular fatty acid presented in ferns only not in the oil? This might help to provide more precise interpretation on the content changes after processing.

Author Response

I am thankful to Reviewers for their comments, questions and corrections. Please find my responses below.

Responses to Reviewer 1

1. In the abstract, authors expressed the content of polyunsaturated fatty acids using diverse units (in mg/g, or in mg/100g) with respect to water content in plant mass. After reading the whole text, it make sense, however after first reading of the abstract only, it sounds a bit confusing. I suggest, though do not insist on the reconsideration using the units in the abstract.

Fatty acid content is now given in mg/g dry weight throughout the abstract.

Abstract: Fern fiddleheads are consumed as a vegetable in many countries. The aim of this study was estimation of three fern species available for sale in the Russian Far East as dietary sources of fatty acids important for human physiology: arachidonic (20:4n-6, ARA), eicosapentaenoic (20:5n-3, EPA) and other valuable long-chain polyunsaturated fatty acids. The content of ARA and EPA was 5.5 and 0.5 mg/g dry weight, respectively, in Pteridium aquilinum, 4.1 and 1.1 in Matteuccia struthiopteris, 2.2 and 0.8 in Osmundastrum asiaticum. Salted fronds of P. aquilinum contained less the fatty acids than the raw fronds: a decrease of up to 49% for ARA and 65% for EPA. Their losses were less pronounced or even insignificant in dried fronds. Cooked ferns preserved significant portions of the long-chain polyunsaturated fatty acids: cooked P. aquilinum contained ARA 4.4 and EPA 0.3 mg/g dry weight (80.1±6.5 and 7.1±0.8 mg/100 g, respectively, in the raw fronds). The ferns may be a supplemental dietary source of these valuable long-chain polyunsaturated fatty acids especially for vegetarian diet.

2. In chapter 2.3 the authors refers to their previous work on method description of identification and quantification of fatty acids. The ref. no 10 however, does not describe what standards were used for quantification of particular fatty acids and where the standards were obtained from. Please complete this information.

The internal standard and its origin (23:0, Sigma) used for quantification of fatty acids is indicated in the first paragraph of section 2.3: “For quantitative analysis of fatty acid content (weight content in plant material), tricosanoic acid (23:0, Sigma) was added as an internal standard before the methanolysis.” The last paragraph of the section was changed accordingly:

Identification and content calculation of fatty acids was performed as described in [10]. Palmitic (16:0) and stearic (18:0) acids used for the calculations of equivalent chain-lengths (ECL) were from Sigma-Aldrich (USA). Quantification of individual fatty acids was based on their FID peak area relative to the internal standard (23:0). No correction factors were used for the calculations of fatty acid content.

3. In chapter Results and discussion, second paragraph, authors names individually those fatty acids that were found in studied raw material in amount equal or higher to 1%. Among them a vaccenic acid (11-octadecanoic acid 18:1n-7) is named. This fatty acid is naturally occurring as far as known in human milk, in ruminant’s fat and in dairy products. Can you discuss the presence of vaccenic acid in plant matrix since it is quite unique observation?

There are two isomers of 18:1n-7: trans and cis. Vaccenic acid is the trans-isomer and, indeed, found in meat and dairy products as a biohydrogenation product of linoleic acid in the rumen of ruminant animals. The cis-isomer (aka cis-vaccenic acid) is a minor component of most plant and animal tissues. It is considered as the elongation product of palmitoleate (16:1n-7) which also occurs in ferns. 18:1n-7 was identified on the basis of ECL and MS-spectrum of DMOX-derivatives. We did not establish the configuration of its double bond (its trans- and cis-isomers have very close retention times by GLC), however on the basis of the literature data, we can assume that it is cis-18:1n-7.

4. The chapter number 3.1 is doubled – please renumber subchapters in Results and Discussion.

Corrected

5. Row 226, the authors cite findings on fatty acid content in ferns from ref. no 20. The results in this article with ref no 20 shows that the highest content was found for linoleic acid (18:2n-6). Out of the sentence in the row 226 it sounds that the highest content in cited ref was found for gamma-linoleic acid (18:3n-6) which is misleading. Please, reformulate the sentence.

Palmitic acid was found to be predominating or second to linoleic acid in young fronds of the fern species in other studies (Etenko et al., 2005; DeLong et al., 2011; Shalisko et al., 2016). The sentence was aimed to stress this. Now it was re-formulated to make it clearer:

Very high content of γ-linolenic acid (18:3n-6) was found for fern fiddleheads in the study [20] which was comparable or even exceeded the content of palmitic acid. This was not observed in our (Table 1) and other studies [17-19].

6. Chapter 3.3 – for better understanding of findings in fatty acid content in cooked ferns, would be reasonable to provide information on used frying oils, the amount and composition. Otherwise, it is not clear on what basis was performed the interpretation and comparison of fatty acid content in cooked and raw ferns. Is there for example any particular fatty acid presented in ferns only not in the oil? This might help to provide more precise interpretation on the content changes after processing.

We used commercially available dishes with cooked ferns. According to the product information of the bracken fern salad, vegetable oil was listed as an ingredient but it was not specified. For the oil amount in the products, it is rather unimportant since we used only fern parts washed and blotted as described in subsection 2.1. and, thus, analyzed only the portion of frying/dressing oil absorbed by the fern tissue. The goal of this part of the study was to estimate preservation of arachidonic and eicosapentaenoic acids in the cooked ferns. Since ARA and EPA present in ferns only and not in the vegetable oil, we compared the absolute content of the fatty acids of interest per dry weight in the cooked and raw ferns.  

Reviewer 2 Report

The authors investigated whether edible ferns grown in the Russian Far East could be a source of dietary LC-PUFA. The results obtained in this study showed that Russian fern plants and their related products could be used as plant-derived LC-PUFA resources.

Overall, the paper seems to be interesting and beneficial. I think there are some improvements that should be made.

  1. Regarding the nomenclature of fatty acids, I am concerned about the mixture of omega- and n-series. How about unifying them into one or the other?
  2. Please include an explanation of the shorthand designation of fatty acids somewhere in the text. e.g. X:Yn-Z fatty acid chain of X carbon atoms and Y methylene-interrupted cis bonds (Z indicates the position of the terminal double bond relative to the methyl end of the molecules)….
  3. Regarding the fern lipids, are the extracted lipids mainly galactolipids and phospholipids? Is it considered that there is almost no triacylglycerol in it?
  4. Tables1-5: How about using the shorthand designation for all fatty acids in the table, instead of using ARA, SCA, etc.?
  5. Tables 2-5: Please format the Tables. For example, "Total lipids, % wet weight" and "Total lipids, % dry weight", please break the line between “Total lipids,” and “% wet weight”. Please center the title in Table 5.
  6. Line 536: “(Engraulis anchoita)” should be “(Engraulis anchoita)”.
  7. Line 365: “52.5 (4.1) for struthiopteris, 40.2 (2.15) for O. asiaticum” should be “52.5 mg/100 g wet weight (4.1 mg/g dry weight) for M. struthiopteris, 40.2 mg/100 g wet weight (2.15 mg/g dry weight) for O. asiaticum”.
  8. Line 367: “7.1±1.9 for Osmunda fern” should be “7.1±1.9 mg/100 g wet weight for Osmunda fern”.
  9. Lines 375-376: “7.1 mg/100 g for aquilinum, 14.1 375 for M. struthiopteris, 14.5 for O. asiaticum” should be “7.1 mg/100 g for P. aquilinum, 14.1 mg/100 g for M. struthiopteris, 14.5 mg/100 g for O. asiaticum”. Is the above-mentioned part 100 g wet weight or 100 g dry weight?
  10. Lines 378-380: “some meat products (29, 28, 27, 378 and 46 mg/100 g edible portion for beef, veal, lamb, and mutton, respectively [33], 21–33 mg/100 g for lamb [36]). Is the above-mentioned part 100 g wet weight or 100 g dry weight?
  11. Lines 381-382: “rabbit (2.9–4.7 mg/100 g [34]), 381 and broiler chicken (1.4–2.4 mg/100 g [37]).” should be “rabbit (2.9–4.7 mg/100 g wet weight [34]), 381 and broiler chicken (1.4–2.4 mg/100 g wet weight [37]).”
  12. Line 387: “18:4n-3 (0.1–0.7)” should be “18:4n-3 (0.1–0.7 mg/g dry weight)”.
  13. Lines 390-391: “fatty acids (5,11,14-20:3 and 5,11,14,17-20:4) are minor components” might be “fatty acids, such as 5,11,14-20:3 (SCA) and 5,11,14,17-20:4 (JA), are minor components”.
  14. Line 546: “D5cis-fatty acids” should be “D5cis-fatty acids”.

Author Response

I am thankful to Reviewers for their comments, questions and corrections. Please find my responses below.

Responses to Reviewer 2

1. Regarding the nomenclature of fatty acids, I am concerned about the mixture of omega- and n-series. How about unifying them into one or the other?

‘Omega’ is commonly used in dietary and health discussions, while ‘n-‘ is more appropriate for chemical designation of fatty acids. Omega-6 and omega-3 were followed by (n-6) and (n-3), respectively, when mentioned for the first time (subsection 3.1., lines 207-208). ‘Omega’ was changed for ‘n’ in the tables and in the text when results are referred to the tables. In subsection 3.4. and section 4. Conclusions, where dietary values of ferns are discussed and highlighted, ‘omega’ was retained.

2. Please include an explanation of the shorthand designation of fatty acids somewhere in the text. e.g. X:Yn-Z fatty acid chain of X carbon atoms and Y methylene-interrupted cis bonds (Z indicates the position of the terminal double bond relative to the methyl end of the molecules)….

The explanation is included in subsection 2.3.:

Shorthand designations of unsaturated fatty acids are as follows: for the fatty acids with methylene-interrupted double bonds in cis configuration, X:Yn-Z indicates a fatty acid chain of X carbon atoms with Y methylene-interrupted double bonds where a terminal double bond is located at Z carbon atom from the methyl terminus (n); a double bond in trans configuration is indicated with “t-“ in the front of the number of carbon atoms; for Δ5-unsaturated polymethylene-interrupted fatty acids (sciadonic and juniperonic acids), the positions of all double bonds are indicated in the front of the numbers of carbon atoms.

3. Regarding the fern lipids, are the extracted lipids mainly galactolipids and phospholipids? Is it considered that there is almost no triacylglycerol in it?

 According to the literature data, there are four major glycerolipid classes in ferns: phospholipids, glycolipids, a betaine lipid DGTS, di- and triacylglycerols. In the mature fronds, triacylglycerols were 12% of total lipids vs. 60% of phospholipids + glycolipids (Robinson et al, 1973) and 0.26 umol/g fresh weight vs. 7 of phospholipids + glycolipids (Sato and Furuya, 1984). In young fronds, it is 7.5-16.9 mole% of total glycerolipids (our unpublished results for New Zealand ferns).

4. Tables1-5: How about using the shorthand designation for all fatty acids in the table, instead of using ARA, SCA, etc.?

 Corrected

5. Tables 2-5: Please format the Tables. For example, "Total lipids, % wet weight" and "Total lipids, % dry weight", please break the line between “Total lipids,” and “% wet weight”. Please center the title in Table 5.

 Corrected

6. Line 536: “(Engraulis anchoita)” should be “(Engraulis anchoita)”.

  Corrected

7. Line 365: “52.5 (4.1) for struthiopteris, 40.2 (2.15) for O. asiaticum” should be “52.5 mg/100 g wet weight (4.1 mg/g dry weight) for M. struthiopteris, 40.2 mg/100 g wet weight (2.15 mg/g dry weight) for O. asiaticum”.

 Corrected

8. Line 367: “7.1±1.9 for Osmunda fern” should be “7.1±1.9 mg/100 g wet weight for Osmunda fern”.

 Corrected

9. Lines 375-376: “7.1 mg/100 g for aquilinum, 14.1 375 for M. struthiopteris, 14.5 for O. asiaticum” should be “7.1 mg/100 g for P. aquilinum, 14.1 mg/100 g for M. struthiopteris, 14.5 mg/100 g for O. asiaticum”. Is the above-mentioned part 100 g wet weight or 100 g dry weight?

 As it is stated in the beginning of the sentence “On the wet weight basis”, the content is in mg/100 g wet weight. Corrected.

10. Lines 378-380: “some meat products (29, 28, 27, 378 and 46 mg/100 g edible portion for beef, veal, lamb, and mutton, respectively [33], 21–33 mg/100 g for lamb [36]). Is the above-mentioned part 100 g wet weight or 100 g dry weight?

 Corrected

11. Lines 381-382: “rabbit (2.9–4.7 mg/100 g [34]), 381 and broiler chicken (1.4–2.4 mg/100 g [37]).” should be “rabbit (2.9–4.7 mg/100 g wet weight [34]), 381 and broiler chicken (1.4–2.4 mg/100 g wet weight [37]).”

 Corrected

12. Line 387: “18:4n-3 (0.1–0.7)” should be “18:4n-3 (0.1–0.7 mg/g dry weight)”.

 Corrected

13. Lines 390-391: “fatty acids (5,11,14-20:3 and 5,11,14,17-20:4) are minor components” might be “fatty acids, such as 5,11,14-20:3 (SCA) and 5,11,14,17-20:4 (JA), are minor components”.

 Corrected

14. Line 546: “D5cis-fatty acids” should be “D5cis-fatty acids”.

Corrected